# Prevention of aspartimide formation during peptide synthesis using cyanosulfurylides as carboxylic acid-protecting groups

Kevin Neumann[1], Jakob Farnung [1], Simon Baldauf[1] & Jeffrey W. Bode [1,2✉]

Although peptide chemistry has made great progress, the frequent occurrence of aspartimide formation during peptide synthesis remains a formidable challenge. Aspartimide formation leads to low yields in addition to costly purification or even inaccessible peptide sequences. Here, we report an alternative approach to address this longstanding challenge of peptide synthesis by utilizing cyanosulfurylides to mask carboxylic acids by a stable C–C bond. These functional groups—formally zwitterionic species—are exceptionally stable to all common manipulations and impart improved solubility during synthesis. Deprotection is readily and rapidly achieved under aqueous conditions with electrophilic halogenating agents via a highly selective C–C bond cleavage reaction. This protecting group is employed for the synthesis of a range of peptides and proteins including teduglutide, ubiquitin, and the low-density lipoprotein class A. This protecting group strategy has the potential to overcome one of the most difficult aspects of modern peptide chemistry.

---

[1] Laboratorium für Organische Chemie, Department of Chemistry and Applied Biosciences, ETH Zürich, 8093 Zürich, Switzerland. [2] Institute of Transformative Bio-Molecules (WPI-ITbM), Nagoya University, Nagoya 464-8602, Japan. ✉email: bode@org.chem.ethz.ch

Advances in organic chemistry have enabled defined, scalable, and cost-efficient synthesis of peptides on solid support as a versatile platform for the reliable preparation of peptides, with particular successes in the discovery of bioactive molecules[1–4]. Improvements such as new coupling agents, dipeptide building blocks, optimized resins, and suppression of racemization allow the production of peptides on a multi-gram scale[5–7]. A plethora of orthogonal-protecting groups facilitates the selective incorporation of fluorophores, drugs, and post-translational modifications[8,9]. These advances, alongside highly efficient ligation techniques such as NCL, KAHA, and Ser/Thr ligation[10–12], enable the routine chemical total synthesis of various proteins[13–15].

Despite the progress made in peptide chemistry, aspartimide formation by cyclization of aspartic acid residue side chains to give cyclic imides remains one of the most formidable obstacles to the synthesis of longer peptide sequences[16]. The base-promoted aspartimide formation occurs during Fmoc removal or peptide coupling (Fig. 1a). The undesired formation of aspartimides during SPPS results in poor yielding or even inaccessible peptide sequences, a problem that—although often encountered—remains unsatisfactorily addressed. It frequently results in costly and time-consuming purification steps in both research and industry[17]. Aspartimide formation is highly sequence-dependent, with glycine, asparagine, aspartic acid, and cysteine in the preceding position showing the highest propensity for aspartimide formation[18].

Three primary approaches to minimize aspartimide formation have been advanced. In the late 1990s, it was reported that a decrease in aspartimide formation is observed by increasing the steric bulk of the aspartic acid ester moiety (e.g., Mbe ester)[19,20]. However, the bulky monomers are expensive and highly hydrophobic, leading to poor coupling efficiency during SPPS. Amide backbone protection with acid-labile groups such as dimethoxybenzyl (Dmb) and 2-hydroxy-4-methoxy-benzyl (Hmb) precludes the possibility of aspartimide formation; however, their coupling efficiency is poor and must therefore be coupled as dipeptides, with only Asp(Dmb/Hmb)Gly being commonly available[21]. Finally, the addition of HOBt and other additives was reported to reduce the amount of aspartimide during SPPS but fails to completely suppress this problem[22,23]. None of the existing strategies for aspartimide prevention provides a general solution.

Here, we report cyanosulfurylide (CSY) as a carboxylic acid-protecting group for the prevention of aspartimide formation. In contrast to ester-based protecting groups, CSY consists of a stable C–C bond that can be selectively cleaved from protected or unprotected peptides with electrophilic halogen species to regenerate the carboxylic acid (Fig. 1b). We demonstrate that it completely suppresses the formation of aspartimide and enhances efficiency during SPPS. This approach improves the synthesis of a highly aspartimide-prone peptide (teduglutide) and enables the synthesis of an otherwise inaccessible peptide (LDLa). In addition, we established that CSY can be selectively removed on a folded protein (ubiquitin), offering a contemporary caging strategy for applications in chemical biology.

## Results

**Reactivity of CSYs.** Our group has previously reported the use of CSYs as a precursor for the synthesis of α-ketoacids for use in KAHA ligation[24,25]. Despite their unusual structure, CSYs show remarkable stability toward the vast majority of reaction conditions including strong acids, strong bases, transition metals, and strong reducing agents[26]. We previously reported that CSYs can be rapidly and chemoselectively oxidized to α-ketoacids by treatment with aqueous, acidic oxone solutions—a process we have used extensively for preparing α-amino acid-derived α-ketoacids for KAHA ligation[25]. Inspired by the exceptional

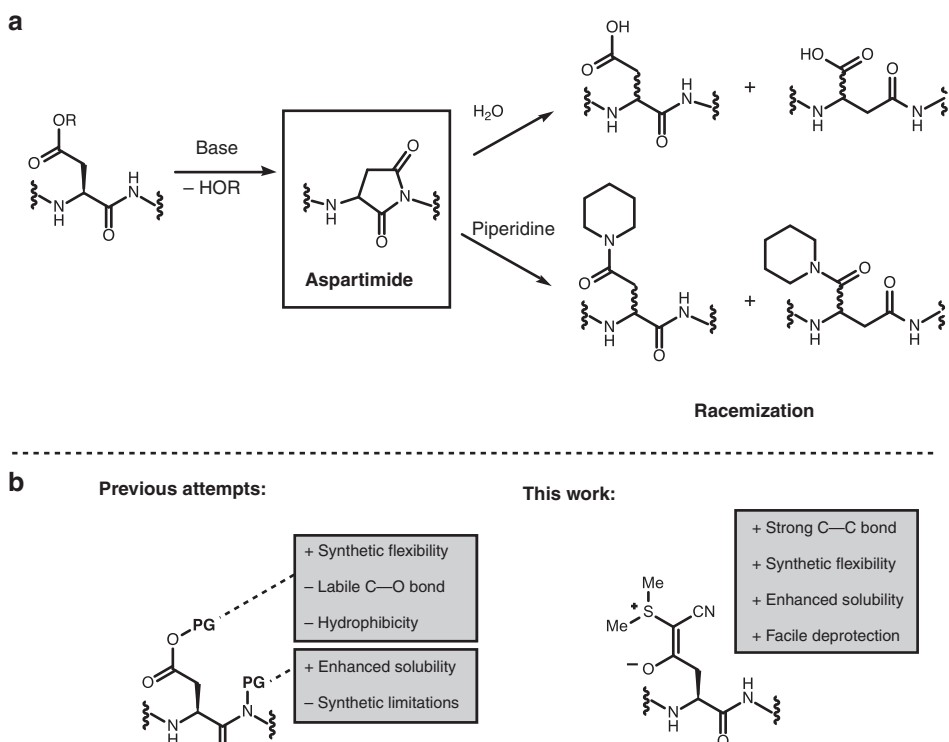

**Fig. 1 Aspartimide formation during SPPS. a** Base-promoted aspartimide formation during SPPS results in racemization and formation of α- and β-peptides. **b** Previous attempts of minimizing aspartimide formation included the usage of bulky esters on aspartic acid side chain and protection of amide backbone. In contrast, this work utilizes a stable C–C bond that masks the side-chain carboxylic acid of aspartic acid.

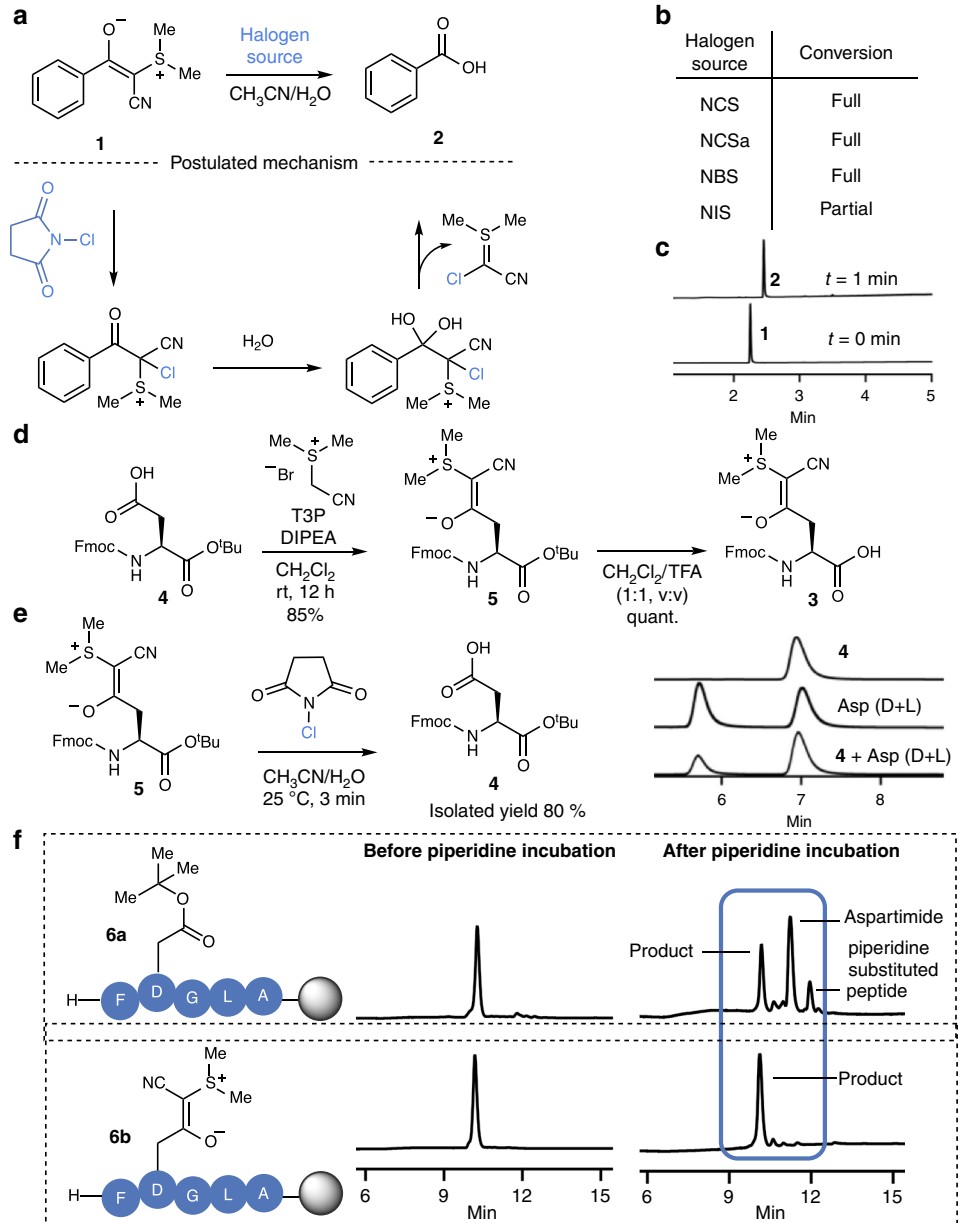

**Fig. 2 Cyanosulfurylides as protecting groups for carboxylic acids. a** Cyanosulfurylide 1 undergoes a rapid reaction with electrophilic halogen species, leading to the release of the unprotected carboxylic acid 2. A postulated mechanism proceeds via electrophilic halogenation followed by hydration and elimination of cyanohalogenylide. **b** Reaction screening was performed using cyanosulfurylide (5 mM) and electrophilic halogen species (10 mM) in aqueous solvent (CH$_3$CN/H$_2$O, 1:1); NCS = N-Chlorosuccinimide, NCSa = N-Chlorosaccharine, NBS = N-Bromosuccinimide, NIS = N-Iodosuccinimide. Reactions were analyzed by LC-MS after 1 min. **c** LC-MS traces of the reaction between sulfurylide 1 and NCS. **d** Fmoc-Asp(CSY)-OH 3 is readily synthesized from commercially available Fmoc-Asp(OH)-O$^t$Bu 4 in two steps. **e** Reaction between aspartic acid monomer 5 and NCS was performed on a larger scale to determine the isolated yield (80%). Chiral HPLC traces showed that the stereoinformation is retained upon CSY removal. **f** Incorporation of cyanosulfurylide-protected aspartic acid into model peptide 6b and comparison with conventional O$^t$Bu ester 6a upon incubation in 20% piperidine in DMF (12 h, room temperature).

stability and ease of handling of CSYs, we sought reaction conditions for their direct transformation to free carboxylic acids, instead of to α-ketoacids. In addition, we required protocols that would be compatible with side chain-unprotected peptides and proteins, allowing us to utilize CSY as a protecting group for aspartic acid during SPPS and peptide ligations.

Based on the postulated mechanism for the conversion of CSY and the related phosphorus ylides, originally developed by Wasserman to vicinal dicarbonyls[27], as well as the documented addition of halogens to sulfurylides[28], we anticipated that we would be able to oxidize CSYs to a species susceptible to C–C

bond cleavage in a manner similar to haloform reactions (Fig. 2a)[29]. With this working hypothesis, we screened various oxidizing agents and observed that electrophilic halogen species rapidly react with CSY 1 to yield the free carboxylic acid 2 under aqueous conditions (Fig. 2b). Among the halogen reagents evaluated, N-chlorosuccinimide (NCS) was particularly promising as it had already been reported to be compatible with all amino acids apart from methionine—which is, however, relatively rare and commonly substituted with norleucine in SPPS[30,31]. The deprotection of CSY 1 proceeded rapidly to full conversion upon addition of NCS under aqueous conditions, providing the free

carboxylic acid **2** (Fig. 2c). These observations are consistent with a mechanism featuring chlorination of the ylide followed by hydration of the carbonyl and loss of the electron deficient ylide species (Fig. 2a).

The carboxylic acids masked as CSYs showed remarkable stability to strongly acidic and basic milieu, as well as to oxidative conditions other than halogenation (e.g., $NaNO_2$ in AcOH), and in the presence of radicals; no major degradation being observed after 1 hour (Supplementary Figs. 1–5, Supplementary Table 1). Having identified conditions to convert CSYs into carboxylic acids in a mild and water compatible reaction, we turned our attention to the stability of CSYs in SPPS, their removal in the context of peptide synthesis, and their effect on aspartimide formation.

**CSY masked aspartic acid for SPPS.** Fmoc-Asp(CSY)-OH **3** was readily prepared from commercially available Fmoc-Asp(OH)-O$^t$Bu **4** in two steps, in 80% overall yield, to provide the bench-stable amino-acid derivative. When Fmoc-Asp(CSY)-O$^t$Bu **5** was treated with an stochiometric amount of NCS, the enantiomerically pure carboxylic acid **4** was obtained in excellent isolated yield (Fig. 2e and Supplementary Fig. 6). We incorporated **3** into the resin-bound pentamer H-FDGLA-OH. Two variants were synthesized; pentamer **6a** was synthesized using traditional Fmoc-Asp(O$^t$Bu)-OH and pentamer **6b** was prepared with Fmoc-Asp(CSY)-OH **3**. With **6b**, we were pleased to observe that no aspartimide formation occurred even after incubation in 20 vol% piperidine in DMF for 12 h at room temperature. In contrast, pentamer **6a**—containing the conventional Asp(O$^t$Bu) monomer—showed a high degree of aspartimide formation and piperidine substituted products (Fig. 2f, for Asp(OMpe) see Supplementary Fig. 7). These results confirmed our hypothesis that masking the carboxylic acid with a C–C bond instead of a C–O ester bond could overcome the problem of aspartimide formation.

We proceeded to identify conditions for the compatible and quantitative deprotection of CSYs on peptides. We synthesized peptide **S2** containing oxidation-sensitive amino acids tryptophan and S$^t$Bu-protected cysteine. Initial attempts to unmask the CSY on-resin using stoichiometric amounts of NCS in DMF/$H_2O$ (9:1, v:v) resulted in a significant amount of aspartimide formation, which we attributed to the highly electrophilic carbonyl that is formed upon chlorination. In contrast, the addition of a small amount of HFIP (DMF/$H_2O$/HFIP (90:8:2)) successfully decreased the amount of aspartimide formed during the deprotection. We were pleased to see that Cys(S$^t$Bu) and Trp (Boc) did not undergo NCS-mediated oxidations under these conditions (Supplementary Fig. 8).

Although cleavage of the CSY groups on resin may be suitable for some peptides, the synthesis of long peptides and proteins would most often benefit from late-stage deprotections of the CSY groups on otherwise unprotected peptide segments. We therefore investigated the removal of the CSY in solution. In contrast to the on-resin deprotection, we were pleased to find that the deprotection of CSY-containing peptide **7** to carboxylic acid peptide **8** was highly selective and no aspartimide formation was observed in acidic $CH_3CN/H_2O$ systems (Fig. 3). We hypothesized that the increased amount of water as well as the more flexible peptide structure in solution precluded aspartimide formation. For deprotection in solution, either buffered or non-buffered aqueous solutions (NaOAc buffer (pH 4.5, 200 mM) or acidic saline (pH 3.0, 200 mM NaCl)) were used and up to 20% $CH_3CN$ was added to ensure solubility of the peptides. After the addition of stochiometric amounts of NCS, the reaction proceeded to completion within 5 min, resulting in the free carboxylic acid containing peptide **8** (Fig. 3). Removal of the

protecting group occurred equally well on purified (after preparative high-performance liquid chromatography; HPLC) and non-purified peptides (after global deprotection). Using these conditions, no formation of N-chloroamines was detected during cleavage of CSY removal (Supplementary Fig. 9).

To demonstrate that no isomerization in form of *iso*-peptide formation occurs during CSY removal, literature known model peptide VYPDGA[32] containing either the free carboxylic acid or CSY were synthesized together with its *iso*-peptide and cyclized aspartimide (peptides **S5** to **S8**, Supplementary Fig. 10). Upon deprotection using stochiometric amounts of NCS, only the deprotected α-peptide was observed. The absence of *iso*-peptide and aspartimide containing peptide indicates that no isomerization occurred upon deprotection.

**CSYs applied for the synthesis of teduglutide.** With these promising results in hand, we sought to apply the CSY-protecting group for the synthesis of biologically relevant molecules. Teduglutide **9**, a 31 amino-acid glucagon-like peptide-2 analog used for the treatment of gastrointestinal diseases[33], possesses two motifs that are prone to aspartimide formation (Fig. 4a)[34]. Indeed, upon synthesizing teduglutide **9** by SPPS using standard conditions, we observed substantial amounts of aspartamide formation. By substituting Asp3 and Asp15 with Asp(CSY) **3**, aspartimide formation was avoided, resulting in a significant increase in yield (Fig. 4b). After purification and isolation of teduglutide(CSY) **10**, the CSY groups were cleaved in either aqueous buffered NaOAc/AcOH (pH 4.5) or in acidic saline (pH 3.0, 200 mM NaCl), both yielding the deprotected peptide **9** in an overall yield of 27% over two steps (vs. 8% utilizing Asp(O$^t$Bu)).

**Synthesis of low-density lipoprotein class A.** Having identified conditions for the quantitative conversion from the CSY to the free acid on fully assembled peptides in solution, we sought to demonstrate the full potential of this protecting group strategy by synthesizing the low-density lipoprotein class A (LDLa) **11**, the N-terminal module (Q23–G63) of the relaxin family peptide receptor 1 (RXFP1) responsible for the receptor activation upon relaxin hormone binding[35]. The LDLa peptide has shown anti-tumorigenic properties, likely owing to its competitive binding to relaxin, which is known to stimulate cancer progression[36]. The synthesis of this 41 aminoacid-long peptide remains a challenge owing to its three aspartic residues, all of which are prone to aspartimide formation (Asp/Asn, Asp/Asp, and Asp/Cys, Fig. 5a). In addition, these residues are located close to the C-terminus, complicating the synthesis owing to repeated exposure to piperidine during deprotection cycles. In previous efforts, this issue was partially circumvented by splitting the synthesis into two fragments that were ligated by native chemical ligation[37]. For the screening of derivatives for medical applications, however, a robust and flexible synthesis of the entire sequence is desirable. In addition to aspartimide formation, the synthesis of LDLa is complicated by the presence of six cysteine residues and its high hydrophobicity. We selected the LDLa module as an excellent target for the evaluation of our developed CSY-protecting group.

Upon synthesis of this peptide with conventional Fmoc-Asp(O$^t$Bu)-OH, a significant amount of aspartimide was observed after coupling 20 amino acids; after the automated sequence was completed, none of the desired product could be detected. Attempts to optimize the synthesis by changing coupling reagents and resin loading did not improve the result. We were pleased to find that simply substituting Fmoc-Asp(O$^t$Bu)-OH with CSY-protected aspartic acid Fmoc-Asp(CSY)-OH **3**, standard SPPS

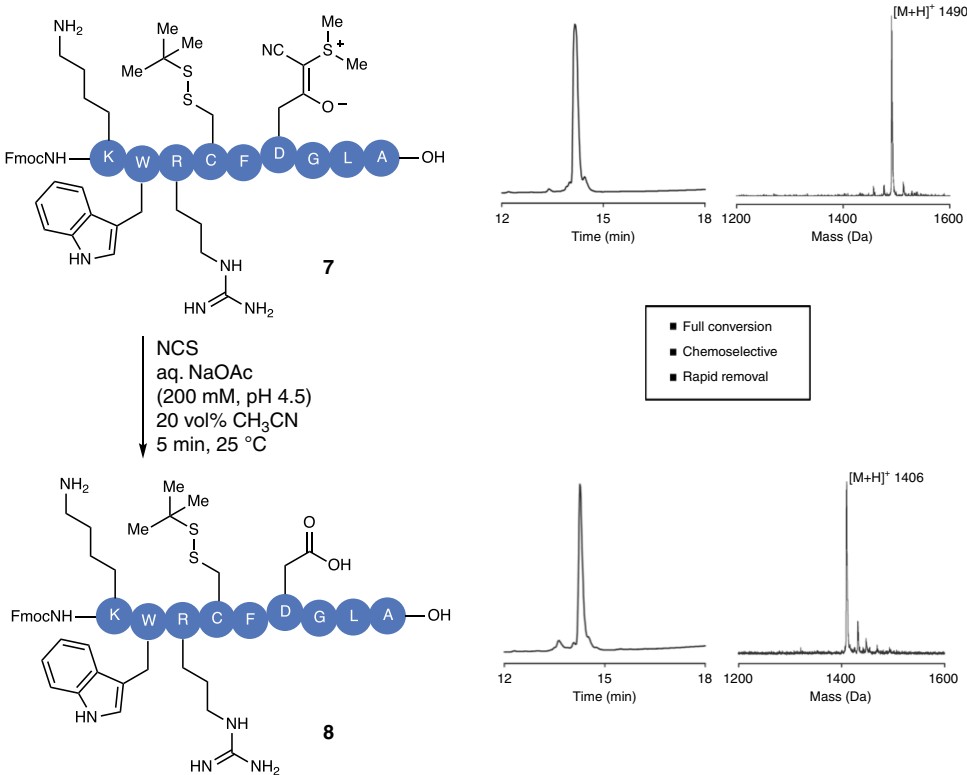

**Fig. 3 Chemoselective removal of cyanosulfurylides.** In solution removal of cyanosulfurylide in the presence of oxidation-sensitive residues including tryptophan and S$^t$Bu protected cysteine. NCS was titrated to the solution until full conversion was indicated by HPLC and mass spectrometry. Reaction went to completion within 5 min at 25 °C without any side-product being observed by HPLC and mass spectroscopy.

conditions provided the desired product LDLa(CSY) **12** as the major peak in the crude HPLC with no aspartimide observed (Fig. 5b).

After cleavage, deprotection of the four CSY residues was readily achieved—in the presence of six Cys(S$^t$Bu) moieties—by titrating a solution of NCS in CH$_3$CN to the peptide to provide LDLa(S$^t$Bu) **13** (Fig. 5c). Finally, the cysteine moieties were deprotected using TCEP under aqueous conditions (pH 7.0, 50 °C, 12 h), delivering unfolded LDLa **14** on a multi-milligram scale. This result demonstrates that CSY-masked aspartic acids can enable the synthesis of otherwise inaccessible peptides. The resulting synthetic LDLa **14** was folded using reported conditions and purified by HPLC (Fig. 5c and Supplementary Fig. 12).

**CSY removal on folded ubiquitin.** We also tested these conditions for late-stage deprotection of folded proteins, as this process could serve as a powerful caging strategy for applications in chemical biology. We synthesized a ubiquitin variant bearing a CSY-utilizing KAHA ligation (Fig. 6). In brief, Asp(CSY) **3** was incorporated into Ub-fragment **15** bearing a photolabile protection group on the N-terminal oxaproline[38–40]. Ub-fragment **15** was purified, deprotected by UV-irradiation giving free hydroxylamine Ub-fragment **16**, and ligated with Ub-fragment **17** containing a C-terminal leucine α-ketoacid, using standard KAHA conditions (HFIP/AcOH, 1:1, v:v, 1 vol% H$_2$O, 22.5 mM); finally, the ligated *depsi*-peptide **18** was rearranged and folded. HPLC and mass spectroscopy indicated that the CSYs are fully compatible with photodeprotection and KAHA ligation conditions. Analysis by mass spectroscopy and HPLC confirmed that CSYs can be selectively removed from folded ubiquitin(E51Hse/D52Asp(CSY)) **19**, resulting in free ubiquitin(E51Hse) **20**. Circular dichroism spectroscopy showed that both ubiquitin

(E51Hse/D52Asp(CSY)) **19** and free ubiquitin(E51Hse) **20** adopted the same secondary structure as recombinant ubiquitin. Both **19** and **20** formed poly-ubiquitin chains in the presence of UBA1, Ube2K, and ATP. The formation of poly-ubiquitin chains is dependent on the correct globular structure of ubiquitin (Supplementary Fig. 13)[41]. These results in conjunction with the CD spectra indicate that at least in the case of ubiquitin, CSY does not impede folding and that CSY can be removed without disruption of the globular structure (Fig. 6c).

## Discussion
We have introduced an alternative approach for preventing aspartimide formation during peptide synthesis by employing stable CSYs as masked aspartic acids. Upon treatment with electrophilic halogen species under aqueous acidic conditions, the ylide is quantitatively converted to the free acid. We have shown that the ylide completely suppresses aspartimide formation during peptide elongation on resin, enabling the synthesis of challenging and aspartimide formation prone peptides. CSYs are readily prepared and—in contrast to Dmb/Hmb backbone protected dipeptides—are not limited to Asp/Gly motifs but can also be used for the protection of other aspartimide-prone motifs such as Asp/Asn, Asp/Cys, and Asp/Asp. Only methionine is not tolerated and needs to be substituted to norleucine. Notably, the ylide as a protecting group exhibits a hydrophilic nature, improving the overall peptide synthesis efficiency. We anticipate that CSYs cannot only be used for prevention of aspartimide formation but also to improve solubility. Owing to its facile synthesis and mild but selective deprotection on both peptides and folded proteins, this concept will be a valuable addition to the efficient synthesis of peptides.

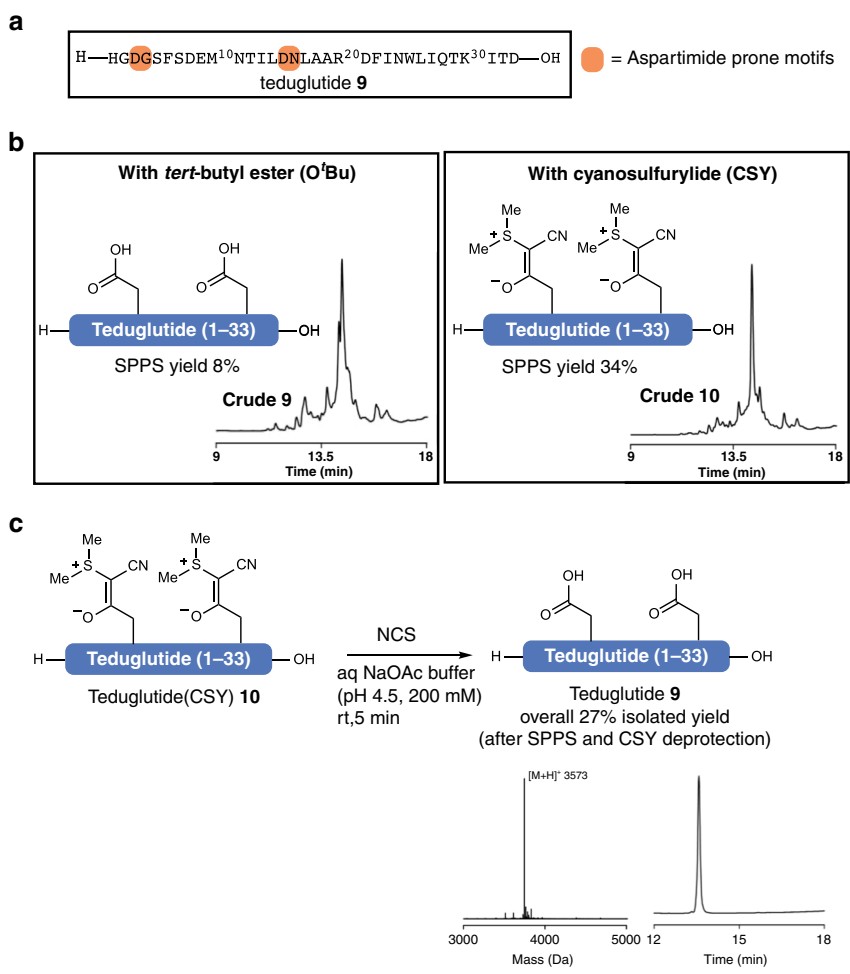

**Fig. 4 Chemical synthesis of teduglutide using CSY. a** Sequence of teduglutide 9 with motifs prone to aspartimide formation highlighted in orange. **b** SPPS synthesis was carried out using standard conditions. Synthesis of teduglutide 9 with conventional tert-butyl ester protected Asp showed significant amount of aspartimide, whereas no aspartimide was detected using cyanosulfurylide Asp(CSY) 3 for the synthesis of teduglutide(CSY) 10. **c** The cyanosulfurylides of teduglutide(CSY) 10 were removed under acidic conditions using NCS.

## Methods

**Removal of CSYs in solution**. Peptides, either purified or after global deprotection, or folded proteins were dissolved in acetic aqueous buffer (e.g., NaOAc, pH 4.5) or acidic saline (e.g., pH 3.0, 400 mM NaCl) with up to 35% CH$_3$CN. Concentration can vary and depends on solubility; better results were observed with an increasing amount of water. When peptides or proteins were fully dissolved, NCS was added from a stock solution in CH$_3$CN (e.g., 100 mM) in portions (Caution: the reaction should be carried out in a well-ventilated hood as the cleaved ylide by product has the potential to degrade to cyanide). The reaction was monitored by analytical HPLC and/or LC-MS (CSYs are absorbing strongly at 254 nm; cleavage results in a decrease in signal). Once full conversion was indicated, the peptides were purified by HPLC; proteins were purified by dialysis.

Best results are obtained if NCS is added in small portions (e.g., 0.5 equiv) to keep the actual concentration of NCS in solution low. Necessary amount of NCS might vary depending on peptide sequence, purity, and concentration.

**Stability and reactivity assays**. For determining the stability and reactivity of CSYs, Fmoc-Asp(CSY)-O$^t$Bu 5 (10 mg, 20 μmol) was dissolved in solvent (100 mM, for solvents see Supplementary Figs. 1–5 and Supplementary Table 1) and the reagent was added to the mixture at room temperature. The reaction was stirred for 1 h and diluted with CH$_3$CN/H$_2$O (1:1, v:v) + 0.1 vol% formic acid (to give a concentration of 1 mM) and analyzed by LC-MS ($\lambda_{abs}$ 220 nm).

**Removal of CSYs on-resin**. After complete peptide elongation on-resin (Rink amide linker-PS resin) loading was determined by Fmoc deprotection. Resin was pre-swelled in CH$_2$Cl$_2$ before incubation in DMF/H$_2$O/HFIP (90:8:2; an increasing amount of water is beneficial for a successful deprotection) with NCS (from 1.5 equiv) for 2 min. The conversion was determined by mass spectroscopy and analytical HPLC. In case of incomplete conversion, the procedure was repeated

with an appropriate amount of NCS. After full conversion, the peptide was cleaved using standard global TFA-mediated deprotection procedure. Depending on the peptide sequence, aspartimide can be formed during on-resin deprotection, thus we recommend doing the deprotection in solution (protocol above).

**Analysis of potential isomerization during deprotection**. Model peptides including CSY protected, CSY unprotected, iso-peptide, and aspartimide peptide S5 to S8 were synthesized and separated on reverse phase HPLC using a slow gradient (5–40% of CH$_3$CN in 30 min, Supplementary Fig. 10). The broad appearance of the peaks is attributed to the slow gradient. CSY was removed from peptide S8 by using stochiometric amounts of NCS in acidic saline (pH 3.0, 400 mM NaCl). After addition of 0.7 equiv of NCS, the deprotected peptide S5-L and some remaining starting material S8 were observed. After additional 0.5 equiv of NCS, only the product S5-L was observed without the presence of any iso-peptide S6 or aspartimide S7. To ensure that iso-peptide S6 and product S5-L are not coeluting, the product was spiked with iso-peptide S6, resulting in two peaks referring to product S5-L and iso-peptide S6. The absence of any iso-peptide and aspartimide confirm that no isomerization occurs during deprotection using NCS.

**Chemical synthesis of teduglutide**. Teduglutide(CSY) 10 was prepared on chloro-trityl PS resin. The resin was loaded with Fmoc-Asp(OtBu)-OH according to the general peptide methods. Amino acid couplings were performed automated on an automated peptide synthesizer (Syro I) using standard Fmoc-SPPS at room temperature with HCTU (4.0 equiv) as coupling reagent and DMF as solvent. Fmoc-Asp(CSY)-OH 3 was coupled manually (2 equiv, 60 min, one coupling). For the peptide cleavage, the peptide was treated with TFA/DODT/H$_2$O (95:2.5:2.5, v/v) for 2 h and the resin was removed by filtration. The solution was concentrated under reduced pressure and triturated with Et$_2$O and centrifuged to obtain crude

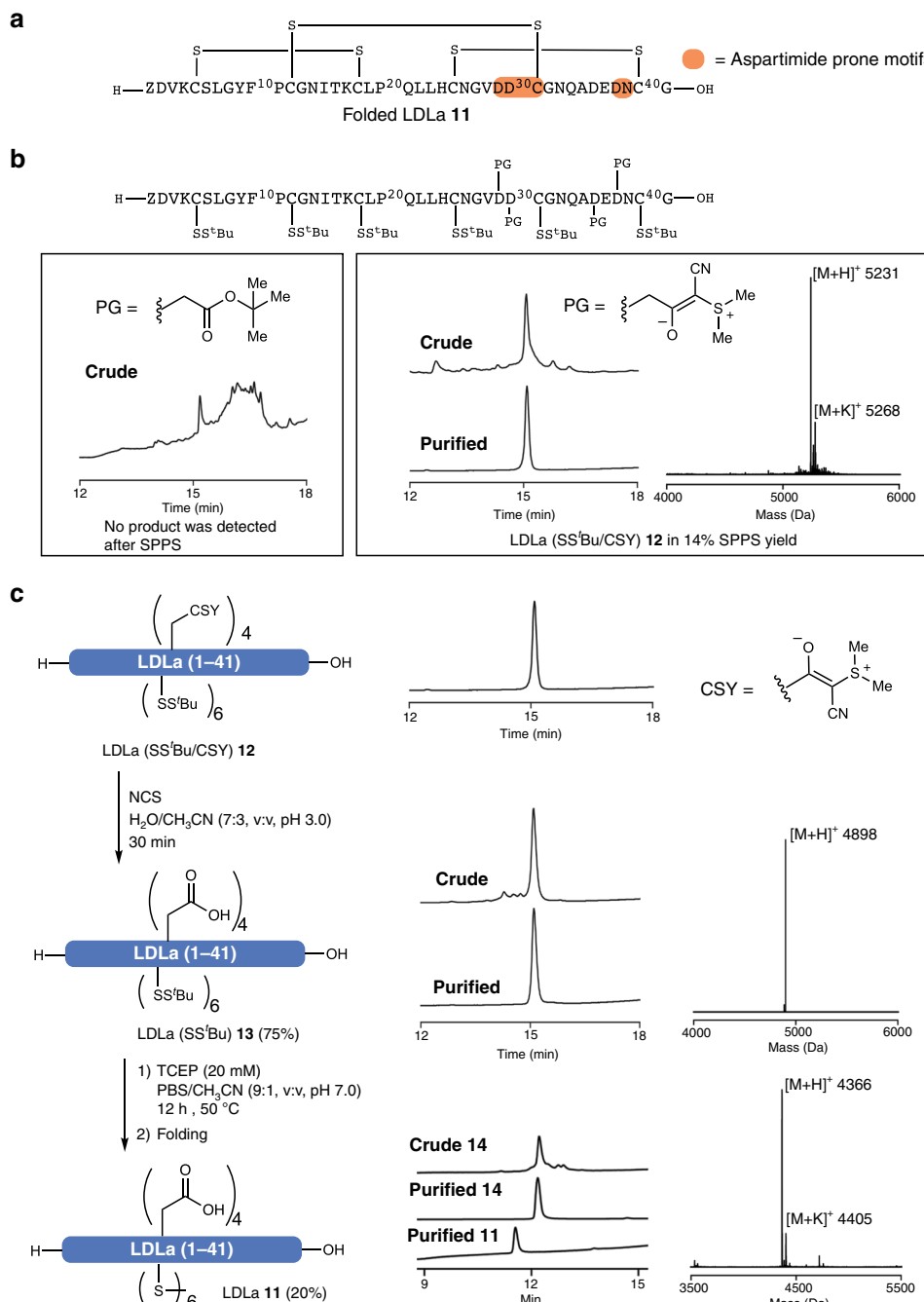

**Fig. 5 Chemical synthesis of LDLa using CSY. a** Sequence of LDLa 11 with sequence motifs that are prone to aspartimide formation are highlighted. Z = pyroglutamic acid. **b** SPPS was carried out using standard conditions. Synthesis with conventional tert-butyl ester protected Asp was not successful, whereas LDLa(S^tBu/CSY) 12 was observed as major product using Fmoc-Asp(CSY)OH 3 monomers. **c** In solution deprotection of cyanosulfurylides using a stoichiometric amount of NCS provides LDLa(S^tBu) 13 in good yield (75%). Cysteines were deprotected using TCEP (20 mM) in PBS/CH₃CN (9:1) giving unfolded LDLa 14 (55%) that was folded to provide LDLa 11 (36%). Detailed folding conditions for 14 are provided in Methods and Supplementary Figs. 11 and 12.

teduglutide(CSY) **10**. The crude peptide was redissolved in H₂O/CH₃CN (1:1, v/v) and purified by preparative HPLC.

Purified teduglutide(CSY) **10** was dissolved in aqueous NaOAc/AcOH (pH 4.5, 200 mM) or acidic saline (pH 3.0, 400 mM NaCl) containing 20% CH₃CN (peptide concentration 1 mM). Once teduglutide(CSY) **10** was fully dissolved, NCS was added from a stock solution in CH₃CN (100 mM) in portions to avoid oxidation of other amino acid moieties (portions of 0.55 equiv). The reaction was monitored by mass spectroscopy and analytical HPLC (note: the CSYs absorb strongly at 254 nm; cleavage results in a decrease in signal). Once full conversion was indicated the crude peptide was purified by HPLC (Method A) and teduglutide **9** was obtained as a white solid after lyophilization.

**Chemical synthesis of LDLa**. Trityl PS resin was loaded with Fmoc-Gly-OH (0.3 mmol/g) using standard procedure. Amino-acid couplings were performed automated on an automated peptide synthesizer (Syro I) using standard Fmoc-SPPS at room temperature with HCTU (4.0 equiv) as coupling reagent and DMF as solvent. Fmoc-Cys(S^tBu)-OH was coupled in absence of base using DIC (1.0 equiv) and Cl-HOBt (1.0 equiv). Fmoc-Asp(CSY)-OH (3.0 equiv) was coupled normally using HCTU (3.0 equiv). After complete elongation of the peptide, L-pyroglutamic acid was coupled with DIC (1.0 equiv) and Cl-HOBt (1.0 equiv). The full peptide was cleaved with TFA/H₂O/DODT (95/2.5/2.5) and purified with preparative HPLC. CSY moieties were removed using NCS (from 1.2 equiv per ylide, stock solution in CH₃CN) in H₂O/CH₃CN (8:2, pH 3.0, 400 mM NaCl), which was

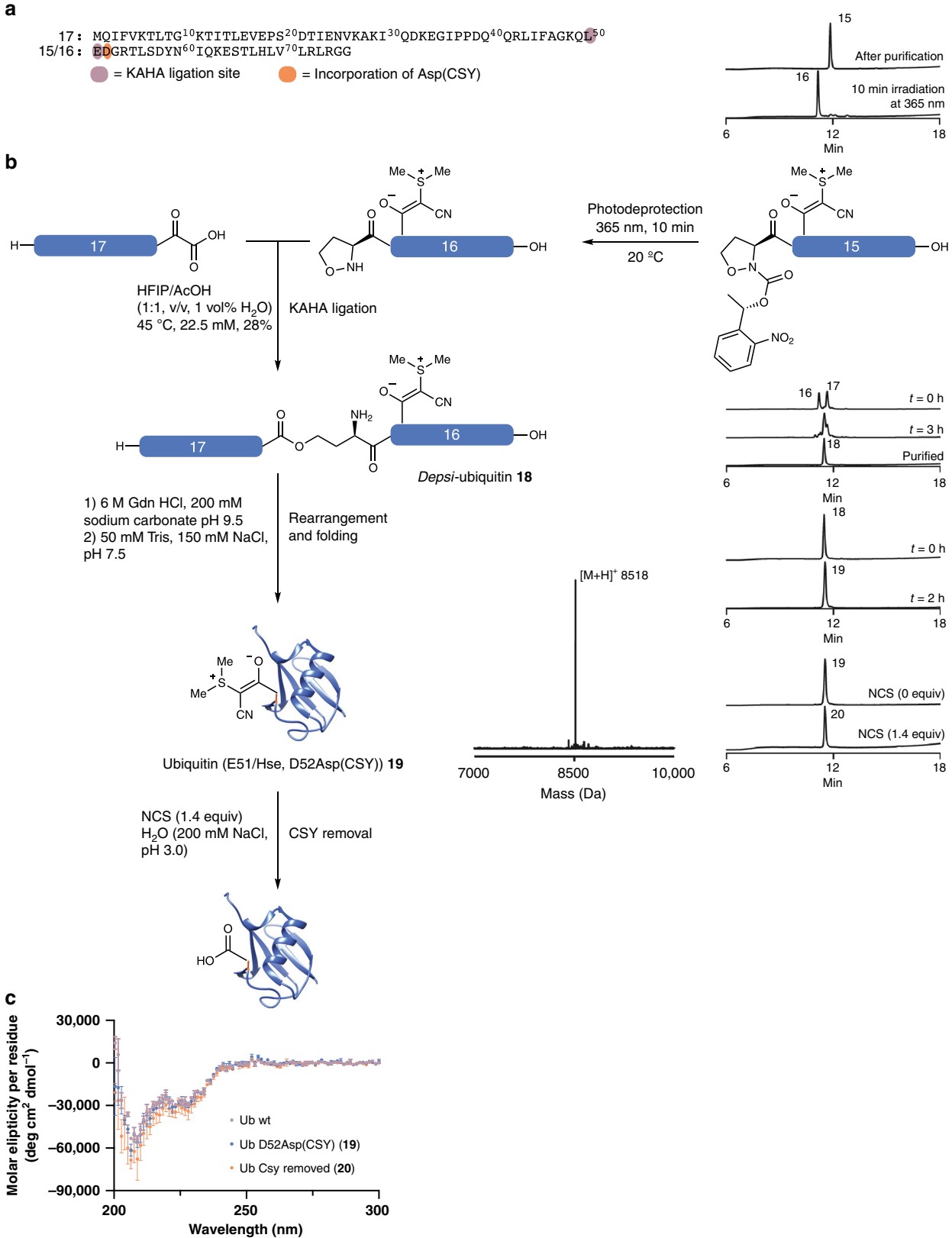

**Fig. 6 Removal of CSY from folded ubiquitin. a** Sequence of ubiquitin with the position of Asp(CSY) and the ligation site highlighted. KAHA ligation was carried out using 16 and 17. **b** Synthesis of ubiquitin (E51Hse/D52Asp(CSY)) 19 and chemoselective removal of CSY on folded ubiquitin(E51Hse/Asp52Asp (CSY)) 19 in acidic saline. **c** CD spectra confirm that the ubiquitin-fold remains with CSY and after CSY removal. Data are shown as the mean of three independent measurements with ±SD. Recombinant ubiquitin is shown for comparison. Ubiquitin structure was reproduced from 5DFL[41].

added in portions. The reaction was monitored via analytical HPLC and mass spectroscopy. After full conversion was observed the peptide was purified either by preparative HPLC or dialysis. The S-*tert*-butyl cysteines were deprotected overnight at 50 °C in PBS/CH₃CN (9:1, pH 7.0, 20 mM TCEP). The final peptide was purified by preparative HPLC and lyophilized.

**Chemical synthesis of ubiquitin-containing CSY**. Ubiquitin segment **15** was prepared on chloro-trityl resin. The resin was loaded with Fmoc-Gly-OH according to the general peptide methods. The automated peptide elongation was carried out on an automated peptide synthesizer (Syro I) according to general peptide synthesis methods. Fmoc-Asp(CSY)-OH **3** was coupled manually (2 equiv, 60 min, one coupling). Fmoc-photoOpr-OH was coupled manually (2 equiv, 180 min, one coupling). For peptide cleavage, the peptide was treated with TFA/DODT/H₂O (95:2.5:2.5, v/v) for 2 h and the resin was removed by filtration. The solution was concentrated under reduced pressure and triturated with Et₂O and centrifuged to obtain crude ubiquitin segment **15**. The crude peptide was redissolved in H₂O/ CH₃CN (1:1, v/v) and purified by preparative HPLC to give photo-ubiquitn **16**. The product containing fractions were irradiated for 10 min at 365 nm. Progress was measured using analytical HPLC. After completion, the solution was lyophilized to give the deprotected peptide.

Ubiquitin Segment **17** was prepared on Rink amide resin. The resin was loaded with Fmoc-Leu-α-ketoacid(acid labile) according to the general peptide methods. The automated peptide elongation was carried out on an automated peptide synthesizer (Syro I) according to general peptide methods. For the peptide cleavage, the peptide was treated with TFA/DODT/H₂O (95:2.5:2.5, v/v) for 2 h and the resin was removed by filtration. The solution was concentrated under reduced pressure and triturated with Et₂O and centrifuged to obtain crude ubiquitin segment **17**. The crude peptide was redissolved in H₂O/CH₃CN (1:1, v/v) and purified by preparative HPLC.

Ubiquitin segment **17** (1.5 equiv) and ubiquitin segment **16** (1 equiv) were mixed with HFIP/AcOH (22.5 μM) at 45 °C and shaken. The progress of the reaction was monitored by analytical HPLC. After 3 h the reaction was deemed complete and diluted with H₂O/CH₃CN (1:1, v/v, 0.1 vol% TFA) and purified by preparative HPLC. The purified depsi-ubiquitin **18** was obtained as a white solid.

Segment **18** was dissolved in rearrangement buffer (200 μL, 6M guanidinium hydrochloride, 200 mM sodium carbonate, pH 9.5) and shaken at room temperature. After 2 h the mixture was given to the folding buffer (10 mL, 50 mM Tris, 150 mM NaCl, pH 7.5) and dialyzed against the folding buffer.

**Ubiquitination assays**. A ubiquitin variant (10 μM), was incubated with Ube2K (2 μM), UBA1 (0.08 μM) and ATP (5 mM) in 50 mM HEPES, 50 mM NaCl, 10 mM MgCl₂, 1 mM DTT, pH 7.5 at 37 °C for 2 h. After 2 h, aliquots were removed and diluted with reducing or non-reducing sodium dodecyl sulfate (SDS) sample buffer. In the case of reducing SDS sample buffer the sample was incubated at 95 °C for 5 min. The samples were resolved using SDS–polyacrylamide gel electrophoresis and visualized using either silver-staining or western blotting against ubiquitin. Anti-Ubiquitin (Cell Signalling, cat. No 3933, 1:1000) and Goat-anti-Rabbit HRP (Cell Signalling, cat. No. 7074 1:5000) were used in western blot analysis (Supplementary Fig. 13).

## Data availability

The data that support the findings of this study are available from the corresponding author upon reasonable request.

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

## Acknowledgements

This study was partially supported by the Swiss National Science Foundation (169451) and ETH Zürich. We thank the LOC NMR Service and Molecular and Biomolecular Analytical Service (MoBiAS) at ETH Zurich for their help.

## Author contributions

K.N. carried out the synthesis and reactivity studies of small molecules and model peptides as well as the synthesis and cleavage study of teduglutide and LDLa derivatives. K.N. contributed to the writing of the manuscript and supporting information. J.F. carried out the synthesis and cleavage studies of ubiquitin derivatives. J.F. contributed to the writing of the manuscript and supporting information. S.B. assisted with the synthesis of small molecules and the writing of the supporting information. J.W.B. and K.N. designed the project. J.W.B. supervised the entire project, assisted in the writing of the manuscript and supporting information.

## Competing interests

The authors declare no competing interests.
