## [Peer Review File · Nature Communications]

Reviewers' comments:

Reviewer #1 (Remarks to the Author):

The manuscript by Neumann and co-workers report their studies on the development of cyanosulfonylides as the aspartic acid protecting to prevent aspartimide formation. Aspartimide formation during the peptide synthesis is often a troublesome problem, although various methods have been developed. None of the strategies for aspartimide prevention provides an general solution. In this report, they developed an interesting protecting group for aspartic acid, cyanosulfonylides (CSY). FMoc-Asp(CYS)-OH can be readily prepared, and readily converted back to the Asp after peptide synthesis. Aspartimide formation during the peptide synthesis was not observed with this building block. Notably, this masked aspartic acid could improve the peptide solubility. With this strategy, they have successfully prepared the peptides which are otherwise difficult to synthesize. Furthermore, they demonstrated that CYS can be used on a folded protein, serving as a new caging strategy. It is highly recommended for publication after addressing the following points:

- 1) Suggest to provide the chiral HPLC spectra to show that compound 4 is enantiomerically pure.
- 2) Figure 4C, there are two mass peaks. What are they?
- 3) Figure 5B, there is a significant mass peak before 5232. What is it?
- 4) page 17, line 14, change "was" to "were"
- 5) in the context, it was mentioned that methionine might not be stable under the oxidation conditions. It is okay for a method to have limitation. Suggest to point out this in the conclusion to alert the researchers who will use this method.

Reviewer #2 (Remarks to the Author):

The authors describe a novel protecting group, that — upon attaching on the side chain of the glutamic acid amino acid — can efficiently suppress the aspartimide formation, a typical side reaction, that often hampers the standard Fmoc solid phase peptide synthesis. The authors thoroughly take into account the already reported strategies that can prevent aspartimide formation in some special cases, and point out that no general solution has been developed yet. The new cyanosulfonylide protecting group — described by the Bode group — is carefully designed by the authors (based on their previous experience on this field) to be easily attachable and efficiently removable from the aspartic acid moiety and still be fully compatible with standard Fmoc chemistry. The authors describe the gram scale synthesis of the cyanosulfonylide protected aspartic acid building block. Furthermore, the conditions for deprotection were carefully examined and optimized for on-resin and for in-solution applications. Bode et. al. demonstrated the utility of the cyanosulfonylide protection group to suppress the aspartimide formation by on the synthesis of two short model peptides. The new findings were applied on the synthesis of the Teduglutide and LDLa peptides that were difficult or impossible to access by the so far reported strategies. The authors also emphasize, the additional feature of their protecting group, namely the zwitterionic structure of the cyanosulfonylide moiety positively contributes to the solubility of the peptides. Finally, the authors demonstrate the compatibility of the cyanosulfonylide protecting group with the alpha-ketoacid hydroxylamine ligation on the synthesis of a ubiquitin variant. The cyanosulfonylide protecting group was successfully removed from the folded protein, indicating the deprotection conditions being very mild.

Significance

Bode et. al. have described the first general approach to prevent aspartimide formation during solid phase peptide synthesis, this approach utilizes cyanosulfonylide protecting group, that is compatible with solid phase peptide synthesis, peptide ligation and can be mildly removed, even from folded proteins. These results might find broad applications in the field of peptide/protein chemistry.

Validity

The authors support their conclusions with solid data.

Data & methodology:

The authors thoroughly examine their findings first on small molecules and model peptides and later apply them successfully on three different proteins. The data provided is appropriate, the experiments were designed thoroughly. Compounds and intermediates analyzed and characterized thoroughly.

References:

The manuscript in part cited the previous literature.

Clarity and context:

The manuscript sufficiently describes the state of art approaches in the field and the utility of the new findings are demonstrated on several examples verifying the expected broad scale applicability of the new strategy.

Questions & Suggestions

- The authors state the N-chlorosuccinimide mediated deprotection only affects the methionine residues. Did the authors observe any cross reactivity between the free epsilon-amino group of the lysine and formation of N-chloroamine?
- In the case of the ubiquitin synthesis the authors have used the photoprotected oxaprolin instead of the more commonly used Boc-oxaprolin. What was the advantage of using the photo protected building block that requires additional deprotection step compared to the Boc variant?
- On page 10, on Figure 4 the MALDI trace shows a significant second peak. Probably it is sodium or potassium adduct of the parent ion, it would be nice if the authors could clarify this.
- On page 12, on Figure 5 B) and C) the authors use inconsistently the "LDLa (StBu/CYS)" vs the "LDLa (SStBu/CYS)" notations, this should be corrected.
- On page 12 of the manuscript and on page 10 of the supporting information the CD spectra would be clearer to see if the x axis (wavelength) would not cross the spectra itself. The x axis should be lowered.
- On page 18 in line 12 the authors write "S-tbutyl" either the "S-tBu" or "S-tertbutyl" notations should be used instead.
- The author should cite previous work by the Kent and the Brik groups on making ubiquitin via two or three ligation which preceded this work although using different ligation approaches.

Reviewer #3 (Remarks to the Author):

The manuscript of Neumann et al. describes a novel aspartic acid derivative for minimizing aspartimide formation during SPPS. The protected aspartic acid residue was also introduced in a protein (ubiquitin), which was assembled from two peptide segments using KAHA ligation. Deprotection was performed using electrophilic halogenating reagents, typically N-chlorosuccinimide (NCS) on the solid phase or in aqueous solution. The authors claim that the method can overcome the difficulty in making aspartimide prone peptides.

I have the following remarks and comments:

1. Asp(OtBu) is certainly not the Asp derivative the authors must use to benchmark their method, because other commercially available Asp derivatives, e.g. Fmoc-Asp(OMpe)-OH, were shown to be better reagents for SPPS than Asp(OtBu).

2. The characterization of the peptides produced in this study is insufficient to enable to conclude on the performance of the reported method.

In Fig.4C, the mass spectrum of the final product 9 is complex and should be explained. I see at least 3 peaks, one being nearly as intense as those of the product.

In Fig. 5C, the mass spectra for 13 and 14 are very complex, the peak attributed to the product being contaminated by many other peaks. Please explain.

In Fig. 6B, the same remark for the MS spectrum of the ubiquitin analog produced.

3. The rationale for preparing Ub by this method is unclear. Wild-type Ub can be easily produced by SPPS or NCL. Who will like to make non natural Ub by this method which require a photochemical deprotection, KAHA ligation of the segments, and two post-ligation treatments, i.e., the rearrangement of the ester bond and the removal of the Asp protecting group? In this work the authors prepared an analog that obviously has an altered secondary structure compared to the wild-type Ub, if we look at Fig. 6C.

4. Regarding Fig 6C, I cannot agree with the authors's claim that the secondary structure of the proteins is similar (note that CD does not allow to conclude on the "globular" structure, as stated by the authors). The authors should repeat CD analyses and present their data as mean +/- SD.

5. The MS spectrum of the Ub analog in Fig. 6B shows a large shoulder > 8500 . Did the authors analyzed the product for the presence of oxidized Met and other side-products?

NCS is used since a while in peptide synthesis and is certainly not considered as a mild chlorination reagent. As mentioned by the authors, NCS can oxidize Met and Cys residues and also induce other side-reactions (see for ex. Selective chemical cleavage of tryptophanyl peptide bonds by oxidative chlorination with N-chlorosuccinimide, Yoram Shechter, Abraham Patchornik, and Yigal Burstein, *Biochemistry* 1976 15 (23), 5071-5075

DOI: 10.1021/bi00668a019.)

In practice and for these reasons, peptide synthesis protocols using NCS are not popular.

Aspartimide formation is a serious problem in the field of peptide and protein synthesis, and novel methods that can help in this area are highly desirable. However, the data provided by the authors in this manuscript do not support their claim.

Point-To-Point Response

Reviewer #1 (Remarks to the Author):

1) Suggest to provide the chiral HPLC spectra to show that compound 4 is enantiomerically pure.

Response: We thank the reviewer for this suggestion and included chiral HPLC traces in Figure 2 showing that Asp(OH) **4** is enantiomerically pure after CSY deprotection. We also compared the chiral HPLC traces to commercially available L- and D-Asp(OH). In total, the data illustrates that no racemization occurs upon CSY deprotection.

2) Figure 4C, there are two mass peaks. What are they?

Response: The two peaks correlate to the M+H and M+K signal. In order to provide less complex mass spectra we performed new mass spectroscopy measurements of the larger peptide fragments and proteins on another mass spectrometer (Synapt G2_Si) and added a desalting step of the sample prior to measurements. The obtained spectra show a significant smaller amount of salt related signals. The mentioned peak that was identified as the sodium-ion signal disappeared.

3) Figure 5B, there is a significant mass peak before 5232. What is it?

Response: We agree with the reviewer that the obtained spectrum of Figure 5B looks too complex – most likely due to salt impurities. As describe above, the new acquired mass spectrum of the same sample (on Synapt G2_Si mass spectrometer) shows less salt related signals and only the required mass of the product.

4) page 17, line 14, change "was" to "were"

Response: The grammatical mistake is corrected.

5) in the context, it was mentioned that methionine might not be stable under the oxidation

conditions. It is ok for a method to have limitation. Suggest to point out this in the conclusion to alert the researchers who will use this method.

Response: We thank the reviewer for the suggestion and incorporated a statement in the conclusion mentioning the incompatibility with Methionine:

“Only methionine is not tolerated and needs to be substituted with norleucine.”

Reviewer #2 (Remarks to the Author):

Questions & Suggestions

- The authors state the N-chlorosuccinimide mediated deprotection only affects the methionine residues. Did the authors observe any cross reactivity between the free epsilon-amino group of the lysine and formation of N-chloroamine?

Response: We thank the reviewer for this interesting point. Indeed, N-chloroamines are readily formed by reactions of amines and chlorinating agents such as NCS. We conducted experiments to determine if N-chloroamines are formed upon addition of NCS under the conditions used for CSY deprotection: Potassium acyltrifluoroborates (KATs) are known to react readily with N-chloroamines at low concentrations with fast kinetics at acidic pH similar to the conditions used for CSY deprotection. We dissolved peptide S3 bearing a lysine and free N-terminus (1 equiv) together with KAT S4 (1 equiv) in acidic saline (pH 3) and titrated NCS to the solution in order to remove the CSY protecting group. Interestingly, after addition of 2 equiv we did not observe any ligation between the peptide S3 and KAT S4. In another experiment we treated peptide S3 (1 equiv) with an excess of NCS (5 equiv) and KAT S4 (15 equiv) in acidic saline (pH 3) and observed the CSY deprotected peptide as the major product; as the only minor byproduct we observed disulfide cleavage and oxidized cysteine moieties, which we attributed to the large excess of NCS. We did not observe any traces of the expected ligation product of peptide and KAT. Both experiments suggest that no N-chloroamines are formed and highlight the affinity of cyanothiopyridones towards chlorination. The experiments indicate that CSY reacts with improved or at least similar kinetics compared to the KAT ligation ($k_2 \approx 10 \text{ M}^{-1} \text{ s}^{-1}$) We included the HPLC of the later experiment in the supporting information.

- In the case of the ubiquitin synthesis the authors have used the photoprotected oxaprolin instead of the more commonly used Boc-oxaprolin. What was the advantage of using the photo protected building block that requires additional deprotection step compared to the Boc variant?

Response: We decided to use photoprotected-oxaprolin instead of Boc-oxaprolin to ease purification of the target peptide. The use of photoprotected-oxaprolin prevents the target peptide from co-eluting with capped species without the need for further purification steps. In addition, the use of photoprotected-oxaprolin enabled us to demonstrate the stability of CSY protected Asp to UV-mediated deprotection conditions.

- On page 10, on Figure 4 the MALDI trace shows a significant second peak. Probably it is sodium or potassium adduct of the parent ion, it would be nice if the authors could clarify this.

Response: We acquired new mass spectra on another mass spectrometer (Synapt G2_Si) including a desalting step prior measurement. Indeed, the sodium ion signal disappeared and only the desired product mass is observed.

- On page 12, on Figure 5 B) and C) the authors use inconsistently the "LDLa (StBu/CYS)" vs the "LDLa (SStBu/CYS)" notations, this should be corrected.

Response: The naming was adjusted to 'LDLa (SStBu/CYS).

- On page 12 of the manuscript and on page 10 of the supporting information the CD spectra would be clearer to see if the x axis (wavelength) would not cross the spectra itself. The x axis should be lowered.

Response: We thank the reviewer for this comment and have changed the CD spectra accordingly.

- On page 18 in line 12 the authors write "S-tbutyl" either the "S-tBu" or "S-tertbutyl" notations should be used instead.

Response: The naming is adjusted to *S-tert-butyl*.

• The author should cite previous work by the Kent and the Brik groups on making ubiquitin via two or three ligation which preceded this work although using different ligation approaches.

Response: We now included the following references of the Kent and Brik group on synthesizing ubiquitin:

- 1) Bang et al *Angewandte Int. Ed.* 2005, 25, 3852.
- 2) Kumar et al *Angewandte Int. Ed.* 2010, 49, 9126.
- 3) Kumar et al *Angewandte Int. Ed.* 2011, 50, 6137.

Reviewer #3 (Remarks to the Author):

The manuscript of Neumann et al. describes a novel aspartic acid derivative for minimizing aspartimide formation during SPPS. The protected aspartic acid residue was also introduced in a protein (ubiquitin), which was assembled from two peptide segments using KAHA ligation.

Deprotection was performed using electrophilic halogenating reagents, typically N-chlorosuccinimide (NCS) on the solid phase or in aqueous solution. The authors claim that the method can overcome the difficulty in making aspartimide prone peptides.

I have the following remarks and comments:

1. Asp(OtBu) is certainly not the Asp derivative the authors must use to benchmark their method, because other commercially available Asp derivatives, e.g. Fmoc-Asp(OMpe)-OH, were shown to be better reagents for SPPS than Asp(OtBu).

Response: We thank the reviewer for identifying this missing experiment. We agree that comparison to Asp(OMpe)-OH would be beneficial and conducted additional experiments.

Besides model peptide **6a** and **6b** containing Asp(OtBu) and Asp(CSY), respectively, we also synthesized model peptide **S1** that was synthesized using Asp(OMpe). It was then incubated in 20% piperidine in DMF in the same way as the other model peptides. As the reviewer suggested a substantial lower amount of aspartimide was observed compared to conventional Asp(OtBu); however, even by using the sterically bulky Mpe ester a significant amount of aspartimide and piperidine substituted peptide is observed. Compared to these two ester derivate, only Asp(CSY) effectively suppresses the formation of aspartimide. We incorporated the HPLC trace of model peptide before and after incubation in 20% piperidine into the supporting information.

2. The characterization of the peptides produced in this study is insufficient to enable to conclude on the performance of the reported method. In Fig. 4C, the mass spectrum of the final product 9 is complex and should be explained. I see at least 3 peaks, one being nearly as intense as those of the product. In Fig. 5C, the mass spectra for 13 and 14 are very complex, the peak attributed to the product being contaminated by many other peaks. Please explain. In Fig. 6B, the same remark for the MS spectrum of the ubiquitin analog produced.

Response: We agree that the obtained mass spectra seem too complex making it hard for the reader to judge the efficiency of the protocol. For that reason, we acquired new mass spectra on another mass spectrometer (Synapt G2_Si) and included a desalting step prior measurement. Indeed, the majority of salt related signals disappeared with only the proton, sodium and potassium signals remaining. We believe the newly acquired mass spectra demonstrate the efficiency of the removal of cyanosulfonyl. The new mass spectra together with HPLC traces and HRMS data should be sufficient for characterization of the reported peptides. All figures are updated with the new mass spectra.

3. The rationale for preparing Ub by this method is unclear. Wild-type Ub can be easily produced by SPPS or NCL. Who will like to make non-natural Ub by this method which require a photochemical deprotection, KAHA ligation of the segments, and two post-ligation treatments, i.e., the rearrangement of the ester bond and the removal of the Asp protecting

group? In this work the authors prepared an analog that obviously has an altered secondary structure compared to the wild-type Ub, if we look at Fig. 6C.

Response: We agree with the reviewer that more straight-forward methods to synthesize ubiquitin are available. We do not suggest that our synthesis of ubiquitin is any way superior to the already published methods. We chose simply to assess the effect of CSY on the folding of a protein and whether CSY can be removed on a folded protein. This particular synthetic route was based on other work in our group.

4. Regarding Fig 6C, I cannot agree with the authors' claim that the secondary structure of the proteins is similar (note that CD does not allow to conclude on the "globular" structure, as stated by the authors). The authors should repeat CD analyses and present their data as mean +/- SD.

Response: We thank the reviewer for the comments regarding the CD analysis. We agree that the phrasing was misleading and have changed it accordingly. We have repeated the experiments and plotted the data according to the reviewer's suggestion. To further support our analysis that the ubiquitin structure is not affected by the incorporation of CSY or its removal we performed a ubiquitination assay. The ubiquitin variants (wt, D52CSY, CSY removed) were incubated with UBA1 and Ube2K in the presence of ATP. Western blot analysis shows the formation ubiquitin-chains. This observation indicates that the ubiquitin-fold remains intact with CSY and after CSY removal as the activation and polymerization of ubiquitin by UBA1 and Ube2K are crucially dependent on the properly folded ubiquitin. Based on the repeated CD measurements and the ubiquitin-chain formation we are confident in our analysis that neither the incorporation of CSY nor its removal with NCS affects the fold of ubiquitin.

*“Both **19** and **20** formed poly-ubiquitin chains in the presence of UBA1, Ube2K and ATP. The formation of poly-ubiquitin chains is dependent on the correct globular structure of ubiquitin (SI S2.7).”*

5. The MS spectrum of the Ub analog in Fig. 6B shows a large shoulder > 8500. Did the authors analyze the product for the presence of oxidized Met and other side-products?

Response: As mentioned above we acquired new mass spectra on another mass spectrometer with an incorporated desalting step and obtained mass spectra showing only the desired product. No oxidation or other side products are observed, which illustrates again the selectivity of the chlorination of cyanosulfurylides. We only observed oxidation of disulfides as a side reaction when excess of NCS was added. As described in the manuscript we substituted methionine with norleucine.

NCS is used since a while in peptide synthesis and is certainly not considered as a mild chlorination reagent. As mentioned by the authors, NCS can oxidize Met and Cys residues and also induce other side-reactions (see for ex. Selective chemical cleavage of tryptophanyl peptide bonds by oxidative chlorination with N-chlorosuccinimide, Yoram Shechter, Abraham Patchornik, and Yigal Burstein, *Biochemistry* 1976 15 (23), 5071-5075, DOI: 10.1021/bi00668a019.)

Response: We agree that NCS can mediate oxidation of several peptide residues as mentioned by the reviewer. However, we developed a protocol in which we keep the amount of free NCS as low as 1 equivalent. We mentioned in the Methods Section that NCS should be added in portions to secure a low concentration of NCS in the solution relatively to the peptide. The reaction proceeds rapidly within couple of minutes at room temperature. This is in strong contrast to other protocols including the reference mentioned by the reviewer (Burstein *Biochemistry* 1976, 15, 5071) in which large excess of NCS (8 equiv) in 50% AcOH is used over a prolonged reaction time. We reassessed our experiments of peptides containing tryptophan residues, and confirmed that we do not observe any oxidation or cleavage in these experiments.

To address the reviewers concern we added a note in the methods that highlights this practical point:

“Note: Best results are obtained if NCS is added in small portions (e.g. 0.5 equiv) to keep the actual concentration of NCS in solution low. Necessary amount of NCS might vary depending on peptide sequence and concentration.”

Reviewers' comments:

Reviewer #1 (Remarks to the Author):

The issues raised by the reviewer previously have been properly addressed.

Reviewer #2 (Remarks to the Author):

Im very happy with the revision made. The paper can be recommended to nat com

Reviewer #3 (Remarks to the Author):

I appreciate the efforts of the authors to provide more data or data of better quality in response to my comments and those of the other referees.

I still consider that the demonstration of the usefulness of CSY for minimizing the aspartimide is incomplete.

As mentioned in Fig. 1A, aspartimide formation is insidious by often representing only the tip of the iceberg of the by-products formed. This is because aspartimide intermediate can be epimerized and opened by water to produce alpha and betapeptide. Because the aspartimide intermediate is prone to epimerization, alpha and betapeptides thus produced are usually epimerized. All these by-products are isobaric of the target peptide and are very complex to separate from it.

Unfortunately, the authors didn't search for the presence of such by-products.

Thus, one crucial experiment is missing. It consists in preparing authentic samples of the by-products that are expected due to aspartimide formation (alpha peptide, epimerized alpha peptide, beta peptide and epimerized beta peptide) and finding chromatographic conditions to separate them.

Then, the analysis of the product obtained using CSY method using the same analytical conditions should clarify if the above mentioned by-products are present or not.

This approach is common in studies dealing with the aspartimide problem. I can recommend the paper of Geiger, T. & Clarke, S. J. Biol. Chem. 262, 785 (1987) which describes model peptides of potential interest for this control experiment.

I recommend that the authors also revise the CD spectra in Fig 6.

A pure alpha helix gives a mean ellipticity per residue at 222 nm of about $-30000 \text{ deg.cm}^2.\text{dmol}^{-1}$. Ubiquitin is not made of α -helix only. These data are therefore inconsistent.

In conclusion, I would be happy to recommend publication of this work if the authors perform the control experiment suggested above and if this experiment clearly show the purity of the end product obtained using CSY chemistry.

Point-to-Point Response

Reviewer 3

- “I appreciate the efforts of the authors to provide more data or data of better quality in response to my comments and those of the other referees. I still consider that the demonstration of the usefulness of CSY for minimizing the aspartimide is incomplete.

As mentioned in Fig. 1A, aspartimide formation is insidious by often representing only the tip of the iceberg of the by-products formed. This is because aspartimide intermediate can be epimerized and opened by water to produce alpha and betapeptide. Because the aspartimide intermediate is prone to epimerization, alpha and betapeptides thus produced are usually epimerized. All these by-products are isobaric of the target peptide and are very complex to separate from it.

Unfortunately, the authors didn't search for the presence of such by-products.

Thus, one crucial experiment is missing. It consists in preparing authentic samples of the by-products that are expected due to aspartimide formation (alpha peptide, epimerized alpha peptide, beta peptide and epimerized beta peptide) and finding chromatographic conditions to separate them. Then, the analysis of the product obtained using CSY method using the same analytical conditions should clarify if the above-mentioned by-products are present or not. This approach is common in studies dealing with the aspartimide problem. I can recommend the paper of Geiger, T. & Clarke, S. J. Biol. Chem. 262, 785 (1987) which describes model peptides of potential interest for this control experiment.”

Response: We thank the reviewer for suggesting this control experiment and agree that this experiment provides more support for the utility of cyanosulfurylides for suppressing aspartimide formation. The suggested literature (J. Biol. Chem. (1987) 262, 785) reports model peptides for the investigation of aspartimide formation and its related byproducts (alpha-, beta-, aspartimide and the respective epimers). We

synthesized these peptide standards using either D-Asp(O^tBu)OH, L-Asp(O^tBu)OH, D-Asp(OH)O^tBu, L-Asp(OH)O^tBu, L-Asp(CSY)OH. In addition, we isolated the corresponding aspartimide product.

By following this report, we were able to separate the alpha- and beta-peptide as well as the aspartimide product on reverse phase HPLC using a slow gradient (Supporting Information, Figure S10). As asked by the reviewer, we removed CSY from the model peptide S5 using stoichiometric amounts of NCS in acidic saline (pH 3). Indeed, we observed a new appearing peak in the HPLC chromatogram corresponding to the alpha-peptide. We did not observe any traces of the beta-peptide or aspartimide. It is reported in the literature (e.g. in the recommended literature) that aspartimide hydrolyzes faster towards the beta-peptide rather than the alpha-peptide. Although we were not able to separate the two diastereomers (D- and L-Asp containing peptides) from each other, the complete absence of beta-peptide and aspartimide is – in our eyes – sufficient to conclude that no isomerization occurs upon CSY removal. The stacked HPLC traces are shown below for your convenience and are included in the supporting information. The poor peak shape is a consequence of the long gradient needed to separate these closely related products.

To reflect these findings, we added the following sentence to the main text:

“To demonstrate that during CSY removal no isomerization in form of *iso*-peptide formation occurs, literature known model peptide VYPDGA containing either the free carboxylic acid or CSY were synthesized together with its *iso*-peptide and cyclized aspartimide (peptides S5 to S8, Supporting Information S2.6). Upon deprotection using stoichiometric amounts of NCS, only the deprotected alpha-peptide was observed. The absence of *iso*-peptide and aspartimide containing peptide indicates that no isomerization occurs upon deprotection.”

A)

alpha-VYPDGA **S5-L**

alpha-VYPDGA **S5-D**

iso-VYPDGA **S6**

Aspl-VYPDGA **S7**

CSY-VYPDGA **S8**

B)

- “I recommend that the authors also revise the CD spectra in Fig 6. A pure alpha helix gives a mean ellipticity per residue at 222 nm of about -30000 deg.cm².dmol⁻¹. Ubiquitin is not made of a-helix only. These data are therefore inconsistent.”

Response: The shown control of recombinant bovine ubiquitin shows the same absorption behavior as the CSY protected and CSY removed ubiquitin variant. Additionally, our CD-spectra show a similar absorption pattern as previously reported for chemically synthesized ubiquitin variants (Angewandte Chemie Int. (2010), 52, 10149).

In conclusion, I would be happy to recommend publication of this work if the authors perform the control experiment suggested above and if this experiment clearly show the purity of the end product obtained using CSY chemistry.

We hope these additional experiments and explanations will conclude this process and look forward to publication of this manuscript.

Kevin Neumann and Jeffrey Bode

REVIEWERS' COMMENTS:

Reviewer #3 (Remarks to the Author):

I acknowledge the authors for having performed the control experiment I suggested. This experiment is of utmost importance and merits to be included in the main manuscript, for example in Fig. 3.

The authors have now provided enough data to justify publication of their work.